# Active Lifestyle Post First Myocardial Infarction: A Comparison between Participants and Non-Participants of a Structured Cardiac Rehabilitation Program

**DOI:** 10.3390/ijerph19063617

**Published:** 2022-03-18

**Authors:** Ran Wainer Shlomo, Rachel Kizony, Menachem Nahir, Liza Grosman-Rimon, Einat Kodesh

**Affiliations:** 1Department of Physical Therapy, Faculty of Social Welfare and Health Sciences, University of Haifa, Haifa 3498838, Israel; lupo70@walla.co.il; 2HaEmek Medical Center, Afula 1834111, Israel; mench@netvision.net.il; 3Department of Occupational Therapy, Faculty of Social Welfare and Health Sciences, University of Haifa, Haifa 3498838, Israel; rkizony@univ.haifa.ac.il; 4Sheba Medical Center, Department of Occupational Therapy, Tel Hashomer, Ramat Gan 52621, Israel; 5Wingate Institute, The Academic College at Wingate, Netanya 4290200, Israel; l.grosman.rimon@gmail.com

**Keywords:** coronary heart disease, myocardial infarction, rehabilitation exercise training, activities

## Abstract

Following myocardial infarction (MI), impaired physical, mental, and cognitive functions can reduce participation in the community and diminish quality of life. This study aims to assess active lifestyle participation and functional performance in patients who were participants and non-participants in cardiac rehabilitation. A total of 71 patients were recruited, 6–10 months after the MI event; 38 chose to participate in a cardiac rehabilitation (CR) program, and 33 did not (NCR). Participation and activity patterns in instrumental activities of daily living, as well as physically demanding leisure activities and social activities, were evaluated using the Activity Card Sort (ACS). Hand grip force and timed up and go (TUG) were tested. A total of 74% of the CR group met physical activity recommendations and only 34% continued to smoke, compared to 39% and 71% in the NCR group, respectively. The CR group, compared to the NCR group, had higher levels of daily activity, social leisure, and physically demanding leisure activities (*p* ≤ 0.001). Null differences between the NCR and CR groups were observed in grip strength and the TUG tests. The study highlighted community participation after MI. Based on a comparison between the groups, the study implies that patients choosing to participate in CR retained higher community participation levels and had better self-management of cardiovascular risk factors.

## 1. Introduction

Myocardial infarction (MI) has an effect on 43% of patients, leading to progressive impairments in health, manifested as a loss of independence and/or physical functioning one year after the event [1]. After MI, patients report severe fatigue, anxiety, depression and concern about their ability to perform daily activities, such as driving, shopping, socialising and returning to work [2,3]. Previous studies demonstrated that a physically active lifestyle has beneficial effects on mortality, morbidity and mental health. Physical activity can reduce the risk of recurrent coronary events and mortality by 59–62% [4], while participation in community activities [5] and maintaining an active lifestyle may prevent heart disease [4] and MI [6]. In addition, participation in leisure activities may play a role as a risk-preventing factor for later-life cognitive decline [7].

A cardiac rehabilitation (CR) program is a multidisciplinary approach, combining physical exercises with medical, social worker and nutritional consultations, developed to bring the patient to optimal levels of physical, mental, and social functioning [8].

For patients, the process of adopting a more active and healthier lifestyle through behavioral changes during CR is difficult and complex. They need to implement several lifestyle changes at once and for some time, as well as struggle against uncertainty and the overprotection of their family members [9]. However, the efficacy of CR programs to maintain short and long term physical activity levels is inconclusive [10].

Regardless of the clear benefits of social participation for wellbeing in cardiac disease, only a few studies have examined the impact of CR programs on participation in daily activities [11,12]. Overall, studies suggest that participants are less restricted in performing household tasks and leisure activities during the rehabilitation period, and their satisfaction with their participation in society and the community improved significantly [13,14]. Most research focuses on specific aspects of participation in daily life, such as returning to work [15] or physical activity [16], and has not compared the level of participation across multiple domains (e.g., leisure, instrumental activities of daily life). Therefore, the objective of the study was to assess daily participation across multiple domains as well as physical functions, between patients who participated and completed versus patients who did not participate in a structured CR program, 6–10 months after their first myocardial infarction.

## 2. Materials and Methods

### 2.1. Study Design

A cross sectional study design was conducted to assess long term participation across multiple domains as well as physical functions, between patients who completed a cardiac rehabilitation program (at least 3 months) versus patients who did not participate in a structured CR program, following their first myocardial infarction. The Ethic Research Board (REB) of Ha’Emek Medical Center approved the study (0180-16-EMC; Clinical Trials; NCT03032146). Informed consent was obtained from each of the participants.

### 2.2. Participants

Seventy-one patients were recruited, 6–10 months after hospitalization for MI, and were divided into two groups based on self-selection into the Structured Cardiac Rehabilitation program (CR group; n = 38) or non-participation (NCR group; n = 33). *Inclusion criteria*: Patients that were hospitalized for the first MI event, who were eligible for CR, and living within 50 km of the rehabilitation center were included in the study. *Exclusion criteria*: An orthopaedic, neurological, or mental disorder that prevented participation in rehabilitation and/or physical activity, according to their medical records.

### 2.3. Procedure

After eligibility was confirmed, a qualified physiotherapist met the participants in their homes. After signing an informed consent form, the participants were characterized regarding their medical history (e.g., previous medical interventions, hypertension, dyslipidemia, diabetes) and their cognitive function with the Montreal Cognitive Assessment—MoCA [17]. Dependent variables were compared between the groups, including lifestyle behavior, functional mobility with the timed up and go (TUG) test, with and without a cognitive task [18], the maximum grip strength for both hands, physical activity level using the short form International Physical Activity Questionnaire (IPAQ) [19,20], and participation in daily activities was measured by the Activity Card Sort (ACS) [21].

### 2.4. Measurements

#### 2.4.1. Participation and Retained Activity Levels

The Activity Card Sort is a self-report measure assessing participation and retained activity levels [21] in daily activities in the domains of instrumental activities (e.g., grocery shopping), physically demanding leisure activities (e.g., swimming) and social activities (e.g., meeting friends). Changes in participation in these activities over time were used to compare and estimate premorbid engagement (before MI) with current activity participation. The outcome measures of the ACS include the number of activities (for each domain and a total score) a person is engaged in at present (fully engaged = 1 point, partially = 0.5 point or stopped doing the activity = 0 points) and the number of activities a person was engaged in before the MI. The “retained activity level” score is reported as a percentage [22]. In addition, participation levels were classified into three categories: retained below 70%, 70–90% and above 90%. The ACS questionnaire is a valid and reliable tool to assess participation in various populations [23,24,25,26].

#### 2.4.2. Physical Activity Level

Assessment was performed using the short form International Physical Activity Questionnaire—IPAQ. It was used to assess walking, medium and heavy physical activity (PA) and the total number of walking and sitting activities throughout the week. The total number of metabolic equivalents (METs) was calculated based on the IPAQ group scoring guideline, weighting each type of activity by its energy requirements, yielding a score in MET-minutes [19,20].

#### 2.4.3. Functional Tests

Timed up and go test (TUG): The TUG was used to evaluate functional mobility, dynamic stability and the risk of falls. Each participant is timed getting up from a chair, walking for 3 m, turning around, returning to the chair and sitting on it again. The subjects performed the TUG test as described above, with and without a cognitive task of subtraction by three (dual task). A time of more than 13.5 s in TUG and a time of more than 15 s in TUG with the dual task, indicates a greater fall risk [27].

Grip strength was measured with a hand dynamometer (JAMAR). Participants were asked to sit comfortably in a chair with back support, with arms fixed and holding the dynamometer. [28] The width of the handle was adjusted to the size of the hand to make sure that the middle phalanx rested on the inner handle. The participant was allowed to perform a one test trial. After this, three trials followed, and the best score was used for the analysis. Handgrip strength was expressed in kilograms.

#### 2.4.4. The Montreal Cognitive Assessment (MoCA)

The MoCA [18] was used to screen cognitive abilities. This measure assesses eight cognitive domains: visuospatial/executive functioning, naming, memory, attention, language, abstraction, delayed recall, and orientation. The highest possible total score is 30 points; a score of 26 or above is considered normal, and scores between 21 and 25 are indicative of mild cognitive impairment (MCI).

### 2.5. Statistical Analysis

SPSS software version 25 was used for analyzing the data. Data are expressed as mean ± SD or median and interquartile range, as appropriate, according to the normal distribution assessed by the Shapiro–Wilks test. The groups were compared with *t*-tests for independent samples or the Mann–Whitney test, depending on the distribution of the data. Chi-squared tests were used for between-group comparisons of physical activity by category, according to the IPAQ and ACS categories of participation. Statistical significance was set at *p* < 0.05.

## 3. Results

Of the 71 study participants, 67 (94%) underwent one of the following interventions: invasive non-surgical coronary angioplasty including a stent (82% of CR, 75% of NCR, *p* = 0.37), balloon insertion (5% of CR, 15% of NCR, *p* = 0.16) or bypass surgery (13% of CR, 3% of NCR, *p* = 0.13). Four subjects (6%) underwent no intervention—one in the CR group and three in the NCR group. The time since the MI was similar between the groups (Table 1). Anthropometric, physical activity level and cognitive function parameters for both groups are presented in Table 1. MoCA and IPAQ scores were significantly higher in the CR group compared to the NCR group (U = 371.5, *p* = 0.02; U = 345.5, *p* = 0.001, respectively).

The IPAQ scores were classified based on the IPAQ guidelines; 74% in the CR group exercised according to WHO recommendations (600–1500 MET/week) compared to only 39% in the NCR group; this distribution was significantly different between the groups (χ^2^
_(2)_ = 9.62, *p* = 0.008).

No statistically significant differences were observed in the functional tests, the grip strength and the TUG tests (Table 1). Fourteen patients (six from the CR group and eight from the NCR group) completed the TUG dual-task test in over 15 s, which is over the threshold for fall detection risk.

Based on the medical records from the time of hospitalization, the most common risk factor was hyperlipidemia, observed in 68% of the participants; hypertension was reported in 58%, and 50% of participants were smokers. One or no risk factors were present in 11% of the CR group compared to 9% of the NCR group. There were no significant differences in the distribution of risk factors between the groups (*p* = 0.39).

There were no statistically significant differences between the groups in the percentage of smokers before the MI (32% in the CR group and 42% in the NCR group, *p* = 0.34). However, 6–10 months after the MI only 11% of the subjects from the CR group still smoked compared to 30% of the NCR group (***χ*^2^** _(1)_ = 4.36, *p* = 0.04).

A total of 78% of the participants returned to work 6–10 months after the MI, while 90% of the NCR group returned to work after the MI (18 of 20 workers) compared to 66% from the CR group (18 of 26 workers) (***χ*^2^** _(1)_ = 39.69, *p* = 0.0001).

### Perceived Level of Participation in Daily Life Activities

ACS participation domains are presented in Table 2. Participants in CR were more engaged in social and physically demanding leisure activities, in the past and present, compared to NCR (social; past: U = 339, *p* = 0.001; present: U = 344.5, *p* = 0.001; physically demanding leisure activities past: U = 294, *p* < 0.001; present U = 340.5, *p* = 0.001).

The results of the retained activity level 6–10 months after the MI, in each domain, in the three categories retained ≥90, 70–90%, and ≤70% of the activities, are presented in Table 3.

A significantly greater number of participants in CR retained more than 90% of social leisure activities, while fewer participants retained less than 70% compared with the NCR group. In the physically demanding leisure domain, a greater percentage of participants in CR retained 70–90% of activities, and less than 70% in a smaller percentage of participants, compared with the NCR group.

## 4. Discussion

The main findings of the current study were that patients opting to participate in a cardiac rehabilitation program after their first MI had higher levels of participation prior to, and 6–10 months after the MI, compared with those who did not participate in a CR program. Moreover, significantly more participants in the CR group maintained a higher percentage of activities in the social and physically demanding domains. Specifically, participants from the CR group incorporated more physical activity and had a healthier lifestyle (i.e., not smoking) than the NCR group. Nevertheless, more participants from the NCR group returned to work after the MI.

Previous studies report that individuals employed at the time of their MI, tend to be enrolled in a CR program earlier to return to work as soon as possible [29,30]. These were not consistent with our results, where 90% of the NCR group returned to work, compared with 69% in the CR group. One possible explanation is that patients who return to work as soon as possible may have less time to participate in the cardiac rehabilitation program. This notion is consistent with prior research reporting that 15.7% of all dropouts from CR programs occur due to returning to work [31]. This finding is important since structured rehabilitation programs are restricted to the operating hours of the cardiac rehabilitation centers, which may not allow working patients to participate in the program. This barrier for participation in the CR program can be addressed by offering other models of CR, such as home-based cardiac rehabilitation.

One of the aims of the CR program is behavior modification to adopt a healthier lifestyle and reduce cardiovascular risk factors [32]. In the current study, this was indicated by the smoking reduction and the higher participation in physical activity of the CR group. The current results show that 74% of the CR group reached the physical activity threshold recommended by WHO (600–1500 MET/week), compared with only 39% of the NCR group, with an average below the recommended levels of only 537 MET/week. These findings are consistent with a recent meta-analysis [33] of the effects of CR programs of up to 12 months post-MI. This meta-analysis showed that the CR groups have higher steps per day and a higher energy expenditure (kcal/week), and a higher proportion of the patients were physically active and less sedentary. These differences in healthier lifestyle behaviors may suggest that participation in a CR program promotes better self-management of cardiovascular risk factors [34,35,36,37].

An active lifestyle, defined as participation in social and physical leisure activities in the community, has a positive effect on the quality of life of many clinical populations [38,39,40]. The return to full social and leisure functioning is one of the CR program objectives [8]. The current results show that patients participating in CR were more active before their MI and 6–10 months after, compared to the NCR group. This is in line with previous studies reporting higher functioning levels to be positively associated with participation in a CR program [41,42].

In this study, the effectiveness of the CR program was evident in the higher percentage of social and physically demanding leisure activities in the CR group. This is similar to previous findings showing that patients who participated in rehabilitation were more engaged in leisure activities [13]. Participation in these two domains of leisure activities (social and physically demanding activities) is pivotal in the prevention of MI recurrence [4] and in the maintenance of quality of life [13].

### Limitations

Active participation in the community following myocardial infarction was examined using a self-reporting questionnaire, in which participants were required to assess their current activity and their activity 6–10 months earlier. In spite of memory bias, all the patients were after their first MI, which is a traumatic event that can constitute a clear reference point. For ethical reasons, a randomized control clinical trial was not conducted. In this observational study, patients were given a choice to participate or to not participate in cardiac rehabilitation. This self-selection into the cardiac rehabilitation program may constitute a potential bias for the effectiveness of CR. The patients who chose to participate in CR may have been more likely to retain long term gains due to pre-existing psychological and behavioral factors than those who chose to not participate. Despite the limitations of the study design, it is important to note that in this study, we captured real-life scenarios of patients who chose and completed CR and those who did not.

Nevertheless, it is possible that factors such as personality traits, which were not assessed in this study, play a significant role in participation in CR. These factors may have important effects on the ability to maintain active lifestyles independently of the CR program.

In this study, we did not perform qualitative analysis, which may contribute to the evaluation of the effect of personality factors on CR program participation. Further, the physical and cognitive functions of the participants before MI, using objective tools, were not measured.

## 5. Conclusions

Six to ten months after MI, the patients who participated in a CR program maintained a high level of social and physical leisure participation and adopted a healthier lifestyle. This study highlights that patients with low levels of participation prior to MI may not choose to participate in CR programs. Therefore, it emphasizes the importance of identifying patients with low participation levels before cardiac events and developing a strategy to encourage them to participate in CR programs. Future randomized controlled prospective studies are required to identify the factors associated with the success of CR programs in maintaining an active lifestyle and participation in multiple daily activity domains.

## Figures and Tables

**Table 1 ijerph-19-03617-t001:** Participant Characteristics.

Variable	NCR Mean ± SD n = 33	CR Mean ± SD n = 38	*p* Value
Gender	7 women 26 men	7 women 31 men	0.77
Age (years)	61.59 ± 11.22	62.75 ± 9.96	0.65
Weight (kg)	86.31 ± 16.02	81.61 ± 12.6	0.18
Height (meters)	1.70 ± 0.1	1.70 ± 0.08	0.67
BMI (kg/m^2^)	29.70 ± 5.24	28.33 ± 3.76	0.22
Dominant Hand Grip (kg)	34.83 ± 11.41	34.47 ± 9.92	0.89
Non-Dominant Hand Grip (kg)	32.30 ± 11.44	32.07 ± 11.12	0.93
	**Median (Min–Max)**	**Median (Min–Max)**	
Months from MI	7 (10–6)	6 (6–10)	0.1
Education (years)	12 (3–20)	12 (0–17)	0.53
MoCA (score)	23 (13–29)	25 (14–30)	0.02
IPAQ (MET/Week)	537 (0–3471)	1158 (99–4132)	0.001
TUG (s)	7.36 (3.19–20.81)	6.68 (4.29–13.13)	0.08
TUG dual task (s)	10.15 (3.04–29.31)	8.94 (3.84–22.90)	
			0.17

BMI—Body Mass Index; MI—Myocardial Infarction; MoCA—Montreal Cognitive Assessment; IPAQ—International Physical Activity Questionnaire; TUG—Timed up and go test.

**Table 2 ijerph-19-03617-t002:** Number of activities (median (min–max)) in participation domains before and after MI.

	NCR n = 33		CR n = 38	
	Past	Present	Past	Present
Instrumental activities of daily living (IADL) (number of activities)	15 (8–18)	14 (2–19)	15 (9–21)	15 (8.5–20.5)
Social leisure activities (number of activities)	11 (5–16)	16 (1–16)	14 * (9–19)	13.5 * (4.5–18.5)
Physically demanding leisure activities (number of activities)	6 (2–13)	3.5 (0–10)	8 * (5–13)	7 * (3–12)
Total (number of activities)	32 (15–43)	30 (3.5–43)	39 * (23–49)	37 * (19–47)

* Indicates significant difference between groups *p* < 0.05.

**Table 3 ijerph-19-03617-t003:** Proportion of retained activity levels per group.

Activity Type	% Retained Participation	NCR (Frequency) % n = 33	CR (Frequency) % n = 38	*p* Value
Instrumental activities of daily living (IADL)	Above 90%	(24) 73	(34) 89	0.06
	70–90%	(4) 12	(3) 8	0.55
	Below 70%	(5) 15	(1) 3	0.07
Social leisure	Above 90%	(23) 70	(34) 89	0.04
	70–90%	(4) 12	(3) 8	0.55
	Below 70%	(6) 18	(1) 3	0.03
Physically demanding leisure	Above 90%	(15) 45	(20) 53	0.70
	70–90%	(5) 15	(15) 39	0.02
	Below 70%	(13) 39	(3) 8	0.004

## Data Availability

The data presented in this study are available on request from the corresponding author.

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
