# Peer review of "Active Lifestyle Post First Myocardial Infarction: A Comparison between Participants and Non-Participants of a Structured Cardiac Rehabilitation Program"

_ijerph, 2022, doi:10.3390/ijerph19063617_

Round 1

Reviewer 1 Report

The approval of the local ethic comittee is missing( not the general Helsinki), I would also like to see an informed consent form.

I suggest to the authors to present in Material and Methods the variables(dependent and independent) and if any correlations to be made, made them more clear. 

From my point of vue, we cannot neglect the bias resulted from differences of personality type, differences of therapy( description for the groups of drugs before and after MI), etc

Did not understood the english translated version of the Limitations, such as "ecological validity". Also, please explain why the study type is a limitation, as long as it could have been interventional also, but it was I suppose the authors choice to be conducted retrospective+observational.

For the Conclusions, they are a rehersal of the limitations plus elusive affirmations like" which are known to have a positive impact on well being and quality of life". Conclusions should reffer only to the results of the present study, which as a matter of fact where too much repeated during the entire paper work.

Author Response

Dear Professor Berrigan,

Section Editor-in-Chief

Re: “Revisions of the manuscript entitled “Active Lifestyle Post First Myocardial Infarction: A Comparison between Participants and Non-Participants of a Structured Cardiac Rehabilitation Program”.

We thank you and the reviewers for their constructive comments and for the opportunity to revise and resubmit our manuscript. We are pleased that the reviewers stated that our study is “original in that it observes a number of domains of activities in the CR program” and “the design is appropriate, the results are clearly presented, and the conclusion and implication is evident”:  We addressed all of the reviewers’ comments and revised the manuscript accordingly.

 Reviewer 1

1.Extensive editing of English language and style required

The manuscript has been edited again for English grammar and style by a native speaker with extensive scientific writing experience

2. The approval of the local ethic committee is missing (not the general Helsinki), I would also like to see an informed consent form”.

We added the following statement: The Ethic Research Board (REB) of Ha'Emek Medical Center approved the study (0180-16-EMC; Clinical Trials; NCT03032146). Informed consent was obtained from each of the participants (Page 2, Line 67).

Please find the attached original consent form (in Hebrew)   

3.I suggest to the authors to present in Material and Methods the variables (dependent and independent) and if any correlations to be made, made them more clear”.

We specified the “dependent variable” list in the revised versions of the manuscript: “Dependent variables were compared between the groups, including lifestyle behavior, functional mobility with the Timed up and go (TUG) test with and without a cognitive task [17]” (Page 2, Line 85).

4. "From my point of vue, we cannot neglect the bias resulted from differences of personality type, differences of therapy (description for the groups of drugs before and after MI), etc"

In agreement with the reviewers, we changed the limitation sections to the following: “It is possible that factors such as personality traits, which were not assessed in this study, play a significant role in participation in CR”. (Page 7, Lines 253-254)

Indeed, differences between therapies may account for the findings.  However, no differences between interventions following MI or distribution of risk factors between groups were found. (pages 3-4, line 146-151)

We added to the results section, the following statement for clarification: “there were no significant differences in the distribution of risk factors between groups (p= 0.39)” (Page 4, Lines 167-168)

 5. “Did not understood the english translated version of the Limitations, such as "ecological validity". Also, please explain why the study type is a limitation, as long as it could have been interventional also, but it was I suppose the authors choice to be conducted retrospective+observational.”

Thank you for this remark. In accordance with the reviewer's comment, we deleted the first sentence from the limitation section, and revised the paragraph as follows: "For ethical reasons, a randomized control clinical trial was not conducted. In this observational study, patients chose to participate in cardiac rehabilitation or not. This self-selection into the cardiac rehabilitation program is a potential bias for the effectiveness of CR. However, it is important to note that in this study we captured real-life scenario of patients who completed CR and those who did not. (Pages 7, Lines 251-258)

6. For the Conclusions, they are a rehersal of the limitations plus elusive affirmations like" which are known to have a positive impact on well being and quality of life". Conclusions should reffer only to the results of the present study, which as a matter of fact where too much repeated during the entire paper work.

In accordance with this comment, we deleted this sentence containing “well-being and quality of life”, which were not assessed in this study. (Pages 7, Lines 279-280)

Thank you again for your insightful comments, which we believe have significantly improved our manuscript.

Sincerely,

Einat Kodesh, Ph.D., Corresponding author.

Reviewer 2 Report

Thank you for this meaningful and important study. As mentioned in the introduction, this study is original in that it observes a number of domains of activities in the CR program. The design is appropriate, the results are clearly presented, and the conclusion and implication is evident. This is a good support for enhancement of CR program as well as advising healthy lifestyle. 

I notice that the study refers to data that is self-reported, although the reliability can be an issue, a better way to make use of this mechanism is to get qualitative feedback as well from the participants. I say this because the paper mentions that the personality factor is not actually considered in this study, and yet it may have an impact on the results. Having qualitative feedback from some, if not all, the participants may help to draw on this factor and allow the study team to evaluate the CR program more thoroughly.

Overall the paper itself is well-written and there is no major issue of clarity or presentation. 

Author Response

Dear Professor Berrigan,

Section Editor-in-Chief

Re: “Revisions of the manuscript entitled “Active Lifestyle Post First Myocardial Infarction: A Comparison between Participants and Non-Participants of a Structured Cardiac Rehabilitation Program”.

We thank you and the reviewers for their constructive comments and for the opportunity to revise and resubmit our manuscript. We are pleased that the reviewers stated that our study is “original in that it observes a number of domains of activities in the CR program” and “the design is appropriate, the results are clearly presented, and the conclusion and implication is evident”:  We addressed all of the reviewers’ comments and revised the manuscript accordingly.

 Reviewer 2

1.“Thank you for this meaningful and important study. As mentioned in the introduction, this study is original in that it observes a number of domains of activities in the CR program. The design is appropriate, the results are clearly presented, and the conclusion and implication is evident. This is a good support for enhancement of CR program as well as advising healthy lifestyle”. 

We are grateful that the reviewer recognizes the value of our study.

2. “I notice that the study refers to data that is self-reported, although the reliability can be an issue, a better way to make use of this mechanism is to get qualitative feedback as well from the participants. I say this because the paper mentions that the personality factor is not actually considered in this study, and yet it may have an impact on the results. Having qualitative feedback from some, if not all, the participants may help to draw on this factor and allow the study team to evaluate the CR program more thoroughly.”

We wish to thank the reviewer for this important comment. Accordingly, we included the following statement in the limitation section: “. In this study, we did not perform qualitative analysis which may contribute to the evaluation of the effect of personality factors on CR program participation.” (Pages 7, Lines 271-272)

 3. Overall the paper itself is well-written and there is no major issue of clarity or presentation.” 

We thank the reviewer for the comment.

Thank you again for your insightful comments, which we believe have significantly improved our manuscript.

Sincerely,

Einat Kodesh, PhD, Corresponding author.

Reviewer 3 Report

The study design says the study was retrospective; however, the description of the participants indicates they were recruited prospectively.  It it also appears that measures were collected only at one time point, thus indicating a cross-sectional design where patients were reporting their perceptions of functioning before and after MI.

It is unclear if the participant self-selected into cardiac rehabilitation group versus the NCR group or if they were assigned to groups.  The time frame of the cardiac rehabilitation is also unclear.  Had all participants completed their rehab or were they still in process?

The authors do not acknowledge the limitation and potential bias of self-selection into cardiac rehabilitation services.  The study indicates that patients who chose cardiac rehab were more active prior to MI and the authors fail to acknowledge that this self-selection could be responsible for the results indicating CR's lasting effectiveness--the patients that chose CR may have just been more likely to engage in physical activity and more likely to change smoking behavior.

Author Response

Dear Professor Berrigan,

Section Editor-in-Chief

Re: “Revisions of the manuscript entitled “Active Lifestyle Post First Myocardial Infarction: A Comparison between Participants and Non-Participants of a Structured Cardiac Rehabilitation Program”.

We thank you and the reviewers for their constructive comments and for the opportunity to revise and resubmit our manuscript. We are pleased that the reviewers stated that our study is “original in that it observes a number of domains of activities in the CR program” and “the design is appropriate, the results are clearly presented, and the conclusion and implication is evident”:  We addressed all of the reviewers’ comments and revised the manuscript accordingly.

 Reviewer 3

1.“The study design says the study was retrospective; however, the description of the participants indicates they were recruited prospectively.  It it also appears that measures were collected only at one time point, thus indicating a cross-sectional design where patients were reporting their perceptions of functioning before and after M”I.

.We agree with the reviewer’s suggestion. We changed the study design type to cross-sectional (page 2 line 63).

2. It is unclear if the participant self-selected into cardiac rehabilitation group versus the NCR group or if they were assigned to groups.  The time frame of the cardiac rehabilitation is also unclear.  Had all participants completed their rehab or were they still

In order to elaborate, we revised the following sentence: “Seventy-one patients 6-10 months after hospitalization for MI, were recruited and were divided into two groups based on self-selection into Structured Cardiac Rehabilitation program (CR group; n=38) or did not (NCR group; n=33)” (page 2 line 72).

We also added the following information “patients who completed cardiac rehabilitation program (at least 3 months)”.  (page 2 lines 64-65)

3. “The authors do not acknowledge the limitation and potential bias of self-selection into cardiac rehabilitation services.  The study indicates that patients who chose cardiac rehab were more active prior to MI and the authors fail to acknowledge that this self-selection could be responsible for the results indicating CR's lasting effectiveness--the patients that chose CR may have just been more likely to engage in physical activity and more likely to change smoking behavior.”

We thank the reviewer for this comment. Indeed, self-selection may introduce a bias. Therefore, we added to the limitation the following: “In this observational study, patients were given a choice to participate or to not participate in cardiac rehabilitation. This self-selection into the cardiac rehabilitation program may constitute a potential bias for the effectiveness of CR.” (page 7 lines 253-256)

Thank you again for your insightful comments, which we believe have significantly improved our manuscript.

Sincerely,

Einat Kodesh, PhD, Corresponding author.

Round 2

Reviewer 1 Report

I thank you for the modifications made, but, I think that erasing some sentences that did not fit in the content doesn t make the content more clear( in refference to points 5 and 6)

Author Response

Re: Revisions of the manuscript titled “Active Lifestyle Post First Myocardial Infarction: A Comparison between Participants and Non-Participants of a Structured Cardiac Rehabilitation Program”.

We thank the reviewers for their constructive and insightful comments, which helped to improve our manuscript.

We addressed all of the reviewers’ comments and revised the manuscript accordingly.

Reviewer 1

Comment 1: “I thank you for the modifications made, but, I think that erasing some sentences that did not fit in the content doesn t make the content more clear (in refference to points 5 and 6)”

Point 5 “Did not understood the english translated version of the Limitations, such as "ecological validity". Also, please explain why the study type is a limitation, as long as it could have been interventional also, but it was I suppose the authors choice to be conducted retrospective+observational.”

We revised this section in accordance with all reviewers’ comments. Instead of using the less familiar term “ecological validity”, we added clarification as follows: “in this study, we captured real-life scenario of patients who completed CR and those who did not” (Page 7, lines 264-266).

We also rephrased and explained that instead of using a randomized controlled clinical trial that is considered a more robust design, we used an observational design, which has limitations. Accordingly, we added a sentence to highlight the potential bias: “Patients who choose to participate in CR may have been more likely to retain long term gains due to pre-existing psychological and behavioral factors than those who chose to not participate” (page 7 , Lines 262-264).

Point 6. "For the Conclusions, they are a rehersal of the limitations plus elusive affirmations like" which are known to have a positive impact on well being and quality of life". Conclusions should reffer only to the results of the present study, which as a matter of fact where too much repeated during the entire paper work."

We deleted the sentence about “well-being and quality of life” since these variables were not addressed in this study, and as suggested by the reviewer, they should not be discussed in the conclusion.

We have now revised this section (Page 8 lines 285-293); however, we believe that some repetition is inevitable in the conclusion section.

"Six to ten months after MI, patients who participated in a CR program maintain high a level of social and physical leisure participation and adopted a healthier lifestyle. This study highlights that patients with low levels of participation prior to MI may not choose to participate in CR programs. Therefore, it emphasizes the importance of identifying patients with low participation levels before cardiac events and developing a strategy to encourage them to participate in CR programs. Future randomized controlled prospective studies are required to identify the factors associated with the success of CR programs in maintaining an active lifestyle and participation in multiple daily activity domains"

Sincerely,

Einat Kodesh, Ph.D., Corresponding author.

Reviewer 3 Report

The authors made adequate changes to the methods section based on reviewer feedback; however, the discussion of limitations remains insufficient.  The authors note that self selection into study groups may bias the study, but they do not provide any discussion on how that bias influences interpretation of the study findings.  The assertion that capturing a real world scenario negates potential selection bias is false.  The real world scenario gives an indication that you are describing patterns of behavior as they are playing out in reality, but it does not do anything to deal with the possibility that those choosing to enroll in CR may have been more likely to retain long term gains due to pre-existing psychological and behavioral factors than those that chose to not do CR.  Without clearly describing the bias, the reader is left with the impression that there is a causal relationship between the rehab and the outcomes.

Author Response

Re: Revisions of the manuscript titled “Active Lifestyle Post First Myocardial Infarction: A Comparison between Participants and Non-Participants of a Structured Cardiac Rehabilitation Program”.

We thank the reviewers for their constructive and insightful comments, which helped to improve our manuscript.

We addressed all of the reviewers’ comments and revised the manuscript accordingly.

Comment 1 "The discussion of limitations remains insufficient.  The authors note that self-selection into study groups may bias the study, but they do not provide any discussion on how that bias influences interpretation of the study findings.  The assertion that capturing a real world scenario negates potential selection bias is false.  The real world scenario gives an indication that you are describing patterns of behavior as they are playing out in reality, but it does not do anything to deal with the possibility that those choosing to enroll in CR may have been more likely to retain long term gains due to pre-existing psychological and behavioral factors than those that chose to not do CR. Without clearly describing the bias, the reader is left with the impression that there is a causal relationship between the rehab and the outcomes"

Thank you for this important comment. We addressed your comment and revised the limitation section as follow:

“Patients who chose to participate in CR may have been more likely to retain long-term gains due to pre-existing psychological and behavioral factors than those who chose to not participate. Despite the limitations of the study design, it is important to note that in this study, we captured real-life scenario of patients who choose and completed CR and those who did not. Nevertheless, it is possible that factors such as personality traits, which were not assessed in this study, play a significant role in participation in CR. These factors may have an important effect on the ability to maintain an active lifestyle independently of the CR program" (Page 7 lines 262-270).

Sincerely,

Einat Kodesh, Ph.D., Corresponding author.